# Comparative Pharmacokinetics and Safety of a Micellar Chrysin–Quercetin–Rutin Formulation: A Randomized Crossover Trial

**DOI:** 10.3390/antiox14111313

**Published:** 2025-10-31

**Authors:** Afoke Ibi, Chuck Chang, Yun Chai Kuo, Yiming Zhang, Peony Do, Min Du, Yoon Seok Roh, Roland Gahler, Mary Hardy, Julia Solnier

**Affiliations:** 1Clinical Research, ISURA, Burnaby, BC V3N 4S9, Canada; aibi@isura.ca (A.I.); cchang@isura.ca (C.C.); yzhang@isura.ca (Y.Z.); pdo@isura.ca (P.D.); mdu@isura.ca (M.D.); kroh@isura.ca (Y.S.R.); 2Factors Group R & D, Burnaby, BC V3N 4S9, Canada; 3Academy of Integrative and Holistic Medicine, San Diego, CA 92037, USA; mary@maryhardy.com

**Keywords:** chrysin, quercetin, rutin, pharmacokinetics, bioavailability, randomized crossover trial, micellar formulation, natural health products, dietary flavonoids, Caco-2 permeability, safety

## Abstract

Chrysin is a dietary flavonoid with antioxidant and anti-inflammatory activity, but its clinical potential is limited by poor oral bioavailability. This randomized double-blind three period crossover trial evaluated the pharmacokinetics of a novel micellar chrysin formulation co-encapsulated with quercetin and rutin (LMC) compared with a non-micellar chrysin formulation (NMC) and unformulated chrysin (UFC). Secondary objectives included in vitro permeability (Caco-2) and a 30-day safety assessment of daily LMC supplementation. Sixteen healthy adults received a single oral dose of each formulation in randomized order separated by a 7-day washout. Plasma chrysin was quantified over 24 h to determine pharmacokinetic parameters. In vitro Caco-2 assays evaluated permeability, and clinical biochemistry of 15 participants were assessed weekly during 30 days of daily LMC use. LMC achieved >2-fold higher systemic exposure than unformulated chrysin (AUC_0–24_ = 914.8 ± 697.5 ng·h/mL; C_max_ = 87.3 ± 59.4 ng/mL; both *p* < 0.05) and >2.6-fold higher than NMC, supported by >10-fold higher in vitro permeability. Daily LMC supplementation was well tolerated, with only mild, reversible adverse events and no clinically relevant safety changes, despite higher systemic exposure. Small, but significant, reductions in fasting glucose were observed in both sexes. The novel micellar chrysin–quercetin–rutin formulation substantially improved bioavailability and was well tolerated during 30 days of daily use, supporting its potential as an advanced delivery strategy for flavonoids with poor oral absorption and identifying glucose regulation as a physiological effect of interest.

## 1. Introduction

Chrysin (5,7-dihydroxyflavone, C_15_H_10_O_4_, molar mass 254.24 g/mol) is a naturally occurring flavonoid primarily found in *Passiflora caerulea* (passionflower), honey, and propolis [1,2,3]. It has garnered increasing attention for its broad spectrum of bioactivities, including anti-inflammatory, antioxidant, anxiolytic, and potential ergogenic effects [1,2,3,4,5]. Its potential antioxidant and mitochondrial effects suggest possible benefits for exercise recovery and cardiovascular fitness [6]. Chrysin (see Figure 1) contains two benzene rings and a pyran ring (a heterocyclic ring containing oxygen). The two rings are linked to its antidiabetic, anti-inflammatory, anticancer, antibacterial, and antiviral activities. Its antioxidant potential is related to the presence of the double bond between C2-C3 and the carbonyl group in the pyran ring [7]. In preclinical studies, chrysin has demonstrated promising pharmacological actions, such as inhibition of aromatase [8], reduction of oxidative stress [9], modulation of inflammatory pathways [10], and protection against various forms of cellular injury [11,12,13,14]. These effects suggest its potential role as a dietary supplement or adjunctive therapeutic agent in conditions ranging from metabolic syndrome to neurodegenerative disorders and athletic performance enhancement [6,15].

Despite its therapeutic potential, the oral application of chrysin in humans has been severely limited by its poor bioavailability. Several factors contribute to its limited systemic exposure after ingestion. First, chrysin is highly lipophilic and exhibits extremely low aqueous solubility (<1 µg/mL), which restricts its dissolution in gastrointestinal fluids and thus limits its absorption across the intestinal epithelium [1,16,17,18]. It is more soluble in organic solvents such as ethanol and DMSO than in water, and its ionization is pH-dependent on its 5- and 7-hydroxyl groups which allow partial ionization under alkaline conditions that can modestly improve solubility [19,20,21]. Second, it undergoes extensive first-pass metabolism in the intestine and liver, particularly via conjugation reactions such as glucuronidation and sulfation. In vitro and in vivo studies have demonstrated that chrysin is rapidly conjugated by UDP-glucuronosyltransferases (UGTs) and sulfotransferases (SULTs), resulting in a predominance of inactive chrysin-glucuronide and chrysin-sulfate metabolites in plasma [22,23]. These metabolic pathways drastically reduce the amount of parent compound available to exert systemic effects.

Efforts to overcome these pharmacokinetic limitations have focused on formulation strategies designed to enhance chrysin’s solubility, protect it from metabolic degradation, and promote its intestinal absorption. Lipid-based delivery systems, such as emulsions, phytosomes, and solid dispersions, aim to improve chrysin’s solubility and permeability through the intestinal mucosa with subsequent lymphatic transport [1,24,25,26,27]. For example, chrysin exhibits weak acidity (pH ≈ 6.7), and its solubility can increase under alkaline conditions due to ionization. However, this approach is limited in vivo because of re-precipitation in acidic compartments [20,28]. Solid dispersion can overcome this pH-dependent solubility limitation by enhancing aqueous solubility and stabilizing drug release across varying pH conditions, mitigating re-precipitation [29]. Nanoparticle formation via co-precipitation with stabilizer can also dramatically enhance solubility by reducing particle size and decreasing crystallinity which also supports improved dissolution independent of pH [30]. Alternatively, micellar delivery systems encapsulate poorly water-soluble compounds in amphiphilic structures that enhance solubilization, increase interaction with enterocytes, and facilitate transcellular absorption [27,31]. These micellar systems may also shield the active compound from enzymatic degradation during gastrointestinal transit [32].

This study presents, for the first time, a novel approach of co-formulating chrysin with the flavonoids quercetin and rutin within a micellar delivery system to address the long-standing challenge of its poor oral bioavailability. Quercetin, a well-characterized dietary polyphenol (see Figure 2a), inhibits key metabolic enzymes, including UDP-glucuronosyltransferases (UGTs), sulfotransferases (SULTs), and cytochrome P450 isoforms, thereby attenuating the first-pass conjugation and efflux of co-delivered flavonoids such as chrysin and improving their absorption [33]. Rutin, the O-glycoside of quercetin (see Figure 2b), further contributes by generating aglycone metabolites through gut microbiota activity, replicating quercetin’s metabolic enzyme inhibition and antioxidant actions, and has been shown to elevate plasma flavonoid levels when administered orally [34]. The co-formulation of these compounds not only capitalizes on their complementary mechanisms of action but also improves their pharmacokinetic profiles. The micellar delivery matrix (LipoMicel^®^) enhances chrysin’s solubility and membrane permeability, while the combined antioxidant and anti-inflammatory activities of quercetin and rutin act synergistically to overcome pharmacokinetic barriers and amplify pharmacodynamic efficacy, thereby increasing the therapeutic potential of chrysin. In addition to chrysin as the primary active compound, the formulation incorporates excipients such as lecithin, and medium-chain triglycerides (MCT) which collectively function to stabilize micelle formation, improve dispersion, and enhance intestinal solubilization and absorption [35,36]. Previous studies on LipoMicel^®^ have shown it to be a highly effective delivery system that has the ability to enhance the absorption of various natural compounds such as quercetin, berberine, omega-3 fatty acids, milk thistle and vitamin D3 [37,38,39,40,41,42]. These formulations have shown enhanced systemic exposure and bioactivity compared to conventional delivery methods, supporting the broader applicability of this delivery system to flavonoids such as chrysin.

While several in vitro and animal studies have demonstrated improved bioavailability of flavonoids using advanced delivery systems, human pharmacokinetic data remains limited, particularly for chrysin. To date, few studies have directly compared multiple chrysin formulations in humans, and the relative bioavailability of micellar chrysin versus conventional oil-based or unformulated preparations remains underexplored. Moreover, most investigations have focused on short-term pharmacokinetic endpoints without characterizing the full 24 h profile or evaluating secondary outcomes such as intestinal permeability and physiological effects following sustained use. Given the unique co-formulation strategy described herein, understanding the pharmacokinetic behavior of this micellar chrysin–quercetin–rutin formulation is critical for rational nutraceutical development. Comprehensive assessment of absorption kinetics (C_max_, T_max_), systemic exposure (AUC), and elimination parameters (T_½_, MRT) provides insight into efficacy potential [16,43], while in vitro Caco-2 cell models—a well-established proxy for human intestinal permeability—offer mechanistic support for formulation-dependent differences [39,44,45]. Beyond pharmacokinetics, there is growing interest in the physiological and safety effects of chrysin supplementation in humans, yet evidence remains scarce. To date, no clinical studies have systematically evaluated the longitudinal effects of chrysin—particularly in combination with synergistic flavonoids such as quercetin and rutin—on key safety and physiological biomarkers in healthy individuals. Such data are essential for establishing both efficacy and safety, especially for individuals using chrysin as part of fitness, wellness, or health optimization regimens.

To address these gaps, this study compared three distinct chrysin formulations—an innovative micellar formulation containing chrysin, quercetin, and rutin; a nonmicellar formulated chrysin preparation; and an unformulated standard chrysin—with respect to in vitro intestinal permeability and in vivo pharmacokinetics. Based on its expected superior absorption profile, only the novel micellar formulation was selected for extended safety evaluation, including serial monitoring of biochemical biomarkers during prolonged use. This comprehensive approach offers, for the first time, direct clinical insight into how advanced formulation technology and synergistic co-flavonoid strategies translate into enhanced bioavailability and verified safety, informing evidence-based nutraceutical development and supporting chrysin’s potential role in human wellness and performance enhancement.

## 2. Materials and Methods

### 2.1. Study Design

The study took place at the ISURA Research Facility during the winter months (December 2024 to March 2025). It was a double-blind, randomized, three-period crossover study designed to evaluate the pharmacokinetics and relative bioavailability of three oral chrysin formulations in healthy adults. Each participant received a single dose of each formulation in random order, separated by a 7-day washout period.

An open-label, single-arm extension phase followed, during which participants consumed a novel micellar chrysin, co-formulated with quercetin and rutin, daily for 30 days to assess safety and physiological effects (Figure 3 and Figure 4; Flowchart).

### 2.2. Participants

Sixteen healthy adults, i.e., healthy men and women between 21 and 65 years of age were enrolled. Inclusion criteria included a voluntarily signed informed consent form, good general health, non-smoking status, and no use of prescription drugs, supplements, or substances known to interfere with flavonoid metabolism. Upon study enrollment, participants completed an online health questionnaire on their medical history. Exclusion criteria included the following:Use of anti-inflammatory or non-steroidal anti-inflammatory (NSAIDs) medication.Presence of cardiovascular diseases and/or other acute or chronic diseases (e.g., liver, kidney or gastrointestinal).Use of cannabis, nicotine or tobacco.Excess drinking of alcohol (>20g/day).Those who were or planned to become pregnant, as well as breastfeeding mothers. Women of childbearing age not using birth control.Use of antioxidant supplements.Use of cholesterol-lowering agents.Concurrent participation in another investigational study.

### 2.3. Ethics and Regulatory Approvals

The study protocol and amendments were reviewed by the Canadian Shield Ethics Review board with OHRP Registration IORG0003491, FDA Registration IRB00004157, and approved on 20 January 2025, under REB Tracking Number: 2022-11-002. The study was conducted in accordance with the ethical principles in the Declaration of Helsinki. The study adhered to CONSORT 2025 guidelines for clinical trials. This study was registered on 15 July 2025, with ClinicalTrials.gov with clinical trial registration number NCT07066839. Written informed consent was obtained from participants upon enrollment.

### 2.4. Formulations and Dosing

Product selection was designed to separate the contributions of LipoMicel^®^ delivery technology and flavonoid co-formulation to chrysin absorption. The novel micellar formulation contained chrysin, quercetin, and rutin in a micelle-based soft gel, while the non-micellar comparator had the same multi-flavonoid composition in a conventional hard gel capsule, enabling evaluation of the micellar system alone. The unformulated chrysin served as a control. Based on absorption performance, only the highest-exposure formulation (micellar chrysin) underwent longitudinal safety evaluation.

The three chrysin-containing interventions examined were as follows (more details in Table 1):Novel Micellar Chrysin: Chrysin, co-formulated with quercetin and rutin as a multi-flavonoid complex, encapsulated in a soft gel with a micelle-based delivery system (LipoMicel^®^; Natural Factors, BC, Canada). From here on written as LMC. More details regarding the LMC Development can be found in Appendix A.Non-micellar Chrysin: Chrysin co-formulated with quercetin and rutin in the same proportions as LMC but encapsulated in a standard hard gel capsule without micellar delivery technology and the required excipients for formulation purposes—enabling evaluation of the micellar system alone. From here on written as NMC.Unformulated/Standard Chrysin: Single ingredient chrysin powder without excipients, encapsulated in hard capsules; served as the baseline single-ingredient preparation, isolating the added effects of co-flavonoid synergy. From here on written as UFC.

### 2.5. In Vitro Caco-2 Permeability Assay

The intestinal permeability of all three chrysin formulations was assessed in vitro using Caco-2 cell monolayers to gain mechanistic insights into their absorption potential at the cellular level. This assay provided predictive data on how each formulation may influence oral bioavailability. The results informed the selection and interpretation of subsequent in vivo pharmacokinetic outcomes.

Caco-2 cells (American Type Culture Collection. HTB-37^TM^. Manassas, VA, USA) were cultured under standard conditions until fully differentiated. Permeability studies were conducted under sink conditions in the apical-to-basolateral direction using previously described methods [39]. Each chrysin formulation was applied to the apical chamber, and transport was measured over a fixed time interval. The apparent permeability coefficient (P_app_, in cm/s) was calculated using the following equation:P_app_ = (dQ/dt)/(A × C_0_)
where dQ/dt is the rate of appearance of chrysin in the basolateral chamber, A is the surface area of the monolayer, and C_0_ is the initial concentration in the apical chamber. Monolayer integrity was confirmed by Lucifer Yellow exclusion.

### 2.6. Bioavailability Study

Following the in vitro permeability assessment, a double-blind, randomized, three-period crossover study was conducted to compare the pharmacokinetics of the three chrysin formulations. Each formulation provided a single oral dose of 1000 mg chrysin, administered with approximately 200 mL of water under fasting conditions, followed by a standardized breakfast 2 h post-dose. A 7-day washout period separated each intervention.

Capillary whole blood samples (50 µL) were collected via fingerstick at pre-dose and at 0.5, 1, 2, 3, 4, 6, 8, 10, 12, and 24 h post-dose using potassium EDTA-coated Minivette^®^ devices (Sarstedt, Germany). Samples were stored at –20 °C until analysis. Plasma chrysin concentrations were quantified by validated HPLC-MS/MS, and pharmacokinetic parameters (AUC_0–24_, C_max_, T_max_, T_½_, MRTi) were calculated.

### 2.7. Safety Study

To extend these findings beyond acute pharmacokinetics, an open-label 30-day extension study was conducted. Participants consumed LMC daily (2 × 500 mg capsules per day; total 1000 mg/day). Safety was monitored through weekly capillary blood sampling (~300 µL) collected in lithium-heparin microtubes and analyzed immediately using an SD-1 Auto Dry Biochemistry Analyzer (Seamaty Technology Co., Ltd., Chengdu, China). Biochemical assessments included liver enzymes (ALT, AST), kidney function (creatinine, eGFR), lipid profile (HDL, LDL, total cholesterol, triglycerides), fasting glucose, and electrolytes. For HbA1c analysis, 10 µL aliquots were lysed in buffer before testing. Participants also completed standardized weekly health questionnaires to report any adverse events or changes in well-being.

### 2.8. Blood Sampling and LC-HRMS Analysis

Sample preparation followed validated protocols [46,47]. Briefly, samples were equilibrated at room temperature and then digested with β-glucuronidase (*Helix pomatia*, ≥100,000 IU, diluted to 330 IU) in pH 5 buffer at 37–40 °C for one hour. Acetone (300 µL) was added for protein precipitation, followed by sonication for one hour and centrifugation at 16,000× *g* for 5 min. The supernatant was transferred to a 96-well plate for LC-HRMS analysis.

Analysis was performed using a Thermo Vanquish UHPLC coupled with a Q Exactive Orbitrap Mass Spectrometer. Separation was achieved on an Acme Xceed C18 column (100 mm × 2.1 mm, 1.9 µm) at 400 µL/min. The mobile phases consisted of 0.5% formic acid in water (A) and methanol (B), with a gradient from 10% to 100% B in 4.0 min.

Quantification was conducted using internal standard calibration with catechin (Sigma, Markham, ON, Canada) (291.0863 *m*/*z*) and a 6-point calibration curve constructed with certified catechin reference standards. Orbitrap settings included a resolution of 70,000 and mass deviation of ±5.0 ppm. Data acquisition and analysis were completed using Thermo Xcalibur 5.0 and TraceFinder 5.0.

The method was validated according to ICH for selectivity, specificity, calibration curve and range (R^2^ = 0.997), accuracy and precision (103% with 4.3% CV), and carryover (0.52%).

### 2.9. Statistical Analysis

Pharmacokinetic parameters were analyzed using repeated-measures ANOVA with post hoc pairwise comparisons (Dunnett correction). Biochemical safety parameters were assessed using repeated-measures ANOVA or mixed-effects models depending on data structure. Cell permeability results were analyzed using repeated-measures ANOVA with post hoc comparisons (Turkey correction). All statistical analyses were conducted using GraphPad Prism (version 10.4), with *p*-values < 0.05 considered statistically significant.

### 2.10. Sample Size and Power Analysis

For the crossover study, sample size was calculated using G*Power 3.1.9.6 based on a within-subjects ANOVA design. Using a large effect size (Cohen’s f = 1.35), α = 0.05, power = 0.95, and ε = 0.75, the minimum required sample size was 4.

For the 30-day safety phase, a paired *t*-test design was used. Assuming an effect size of 1.0, α = 0.05, and power = 0.80, the required sample size was 10. Reference values and variability from prior micellar flavonoid studies were used to model effect sizes for ALT, AST, and sodium levels.

### 2.11. Randomization and Blinding

Simple randomization was performed using RAND() function in Microsoft Excel to allocate participants to one of six treatment sequences with 1:1:1 ratio. An independent study assistant, who was not involved in data collection or analysis, managed treatment assignment and capsule dispensing. All capsules were placed in identical opaque containers labeled only with participant IDs. The study coordinator, participants, and principal investigator were blinded to the treatment allocation throughout the crossover phase.

### 2.12. Adverse Event Reporting

Participants were instructed to complete weekly health questionnaires and report any adverse events (AEs) to the study team. AEs were categorized based on severity and clinical relevance, according to criteria detailed in Appendix A.

### 2.13. Formulation Characterization

To provide mechanistic context for the pharmacokinetic and permeability results, key physicochemical properties of the three formulations were characterized under standardized conditions. Apparent solubilized concentrations of chrysin and co-flavonoids were quantified in simulated gastric fluid (SGF) and simulated intestinal fluid (SIF), and dispersion behavior was evaluated by laser diffraction particle size analysis. Full experimental protocols and detailed characterization results are provided in Appendix A.

## 3. Results

### 3.1. Demographic and Baseline Characteristics

A total of 18 healthy adults (7 males, 11 females; mean age 37.2 ± 10.2 years) were enrolled in the pharmacokinetics and safety study. 16 of the adults participated and completed the pharmacokinetic study. 15 adults participated in the safety study. Baseline demographic and clinical characteristics were recorded at week 0 prior to the first intervention and are summarized in Table 2.

### 3.2. Pharmacokinetic Profiles

Following administration, plasma concentrations of chrysin differed markedly among the three formulations (Figure 5). All formulations demonstrated a gradual increase in plasma chrysin concentration, peaking around 4–6 h post-dose. LMC achieved markedly higher plasma concentrations than NMC and UFC. As shown in Table 3, both overall exposure (AUC_0–24_) and C_max_ were significantly (*p* = 0.0221 and *p* = 0.0128) higher for LMC (AUC_0–24_ 914.8 ± 697.5 ng·mL^−1^·h; C_max_ 87.3 ± 59.4 ng/mL) compared to NMC (AUC_0–24_ 345.7± 280.8 ng·mL^−1^·h; C_max_ 32.9 ± 23.5 ng/mL) and UFC (AUC_0–24_ 456.9 ± 356.4 ng·mL^−1^·h; C_max_ 42.1 ± 28.4 ng/mL). No statistically significant differences were observed in T_1/2_ (LMC: 25.3 ± 59.1 h; SDT: 9.0 ± 3.3 h; UFC: 11.0 ± 3.5 h; *p* = 0.6963), T_max_ (LMC: 4.8 ± 3.8 h; SDT: 5.8 ± 3.9 h; UFC: 5.8 ± 5.7 h; *p* = 0.4903), or MRTi (LMC: 38.4 ± 85.9 h; SDT: 13.3 ± 4.2 h; UFC: 19.1 ± 8.4 h; *p* = 0.4207) between formulations.

### 3.3. Caco-2 Permeability

The apparent permeability coefficients (P_app_) of chrysin across Caco-2 cell monolayers differed significantly among the three tested formulations (Figure 6). LMC demonstrated the highest permeability, with a mean P_app_ of 3.08 × 10^−5^ cm/s (range: 2.46–3.73 × 10^−5^ cm/s). NMC showed intermediate permeability, with a mean P_app_ of 8.89 × 10^−6^ cm/s (range: 5.41 × 10^−6^–1.48 × 10^−5^ cm/s). UFC exhibited the lowest permeability, with a mean P_app_ of 2.22 × 10^−6^ cm/s (range: 1.02–3.35 × 10^−6^ cm/s). The LMC formulation showed a >13-fold increase in permeability compared to UFC and was significantly higher than both NMC and UFC (*p* < 0.001). These results support the enhanced intestinal transport potential of the micellar delivery system and are consistent with the in vivo pharmacokinetic findings.

### 3.4. 30-Day Safety Outcomes

Daily supplementation with LMC (1000 mg/day) for 30 days was well tolerated. No clinically significant changes were observed in liver enzymes or renal function markers at any time point (Table 4 and Table 5).

Among male participants, repeated-measures ANOVA identified significant time effects for fasting glucose (*p* = 0.0369), total CO_2_ (*p* = 0.0118), and sodium (*p* = 0.0061). Post hoc analysis showed a modest increase in total CO_2_ and a decrease in serum sodium at day 30 (*p* < 0.05 and *p* < 0.01, respectively). Mean sodium levels remained slightly below reference in some individuals (mean: 133.8 mmol/L). However, considering that sodium concentrations in capillary blood are typically 2–3 mmol/L lower than in venous samples, these values likely remain within the physiological reference range [46]. Reductions in fasting glucose over time were observed in both sexes, with values remaining within normal clinical ranges. Among female participants, most biochemical parameters remained stable. A significant reduction in fasting glucose was observed from baseline to day 30 (mean ± SD: 5.2 ± 0.6 vs. 4.2 ± 0.5 mmol/L, *p* < 0.01; *p* for time = 0.0009). No other significant changes were detected in safety parameters. Together, these findings support the short-term safety of LMC in healthy adults.

### 3.5. Adverse Events Report

LMC was generally safe and well tolerated over the 4-week supplementation period, with a low incidence of mild adverse events and no serious outcomes. One participant reported mild bloating, heartburn, and abdominal discomfort throughout the study. These symptoms improved when the supplement was taken with food and did not require discontinuation. Another participant developed a pruritic rash and generalized itchiness after Week 2, which led to early discontinuation. Symptoms resolved following cessation of supplementation; it remains unclear whether the rash was directly related to chrysin supplementation, food intake, or other unrelated factors (e.g., environmental exposures).

A summary of all reported adverse events is provided in Appendix A.

### 3.6. Micellar Formulation In-Vitro Characterization

Key physicochemical properties of the three formulations are presented in Appendix A. The micellar formulation (LMC) showed substantially higher apparent solubilized concentrations of chrysin and co-flavonoids in simulated gastric and intestinal fluids compared with the non-micellar and unformulated controls and exhibited a more favorable dispersion profile as determined by particle size analysis. These characteristics paralleled the clinical pharmacokinetic findings, with greater solubilization and improved dispersion associated with higher systemic exposure (AUC and Cmax).

## 4. Discussion

This study is the first to directly evaluate a novel micellar chrysin formulation co-containing quercetin and rutin (LMC) against two reference comparators, providing an integrated assessment of pharmacokinetics, intestinal permeability, and short-term safety in healthy adults. The LMC formulation demonstrated markedly enhanced oral bioavailability compared with both the non-micellar multi-flavonoid formulation (NMC) and the unformulated chrysin control (UFC), underscoring the combined benefits of advanced delivery technology and synergistic flavonoid co-formulation. Mechanistic in vitro studies using Caco-2 cell monolayers supported these findings, revealing increased intestinal permeability consistent with the observed improvements in systemic exposure. Furthermore, daily LMC supplementation for 30 days was well tolerated and produced no clinically meaningful changes in safety biomarkers, supporting the physiological safety of higher systemic chrysin exposure achieved via this novel formulation. Collectively, these results establish proof of concept that rational formulation strategies—combining micellar technology and metabolic synergy through co-flavonoids—can overcome inherent bioavailability limitations of chrysin while maintaining a favorable safety profile.

### 4.1. Enhanced Bioavailability Through Micellar Delivery

Native chrysin is known for its extremely low oral bioavailability, estimated at less than 1% and reported in the range of 0.003–0.02% [1,9]. This is primarily due to its poor aqueous solubility (~0.1 g/L), rapid metabolism via glucuronidation and sulfation, and active efflux by intestinal transporters such as MRP2 and BCRP [3,22,47,48,49]. Prior human studies administering 400 mg of unformulated chrysin reported peak plasma levels of only 3–16 ng/mL and AUC values between 5 and 193 ng/mL·h [22]. In the current study, even unformulated chrysin showed improved systemic exposure compared to previous literature—possibly due to the higher dose (1000 mg) used.

LMC achieved the highest systemic exposure, with a mean AUC_0–24_ of 914.8 ± 697.5 ng/mL·h, representing a 2.6-fold increase compared with NMC (345.7 ± 280.8 ng/mL·h) and a 2.0-fold increase compared with UFC (456.9 ± 356.4 ng/mL·h). C_max_ was also significantly greater for LMC (87.3 ± 59.4 ng/mL) compared with both NMC (32.9 ± 23.5 ng/mL) and UFC (42.1 ± 28.4 ng/mL; *p* = 0.0128), indicating improved absorption efficiency. T_max_ values were similar across treatments, suggesting that micellar encapsulation primarily increased the extent rather than the rate of absorption. These results demonstrate that micellar delivery technology markedly enhances chrysin bioavailability beyond the modest contribution of quercetin and rutin co-formulation alone, while also suggesting a potential additive role of flavonoid synergy (NMC vs. UFC). In fact, when compared to unformulated chrysin, co-formulation alone was able to achieve an increase in permeability (albeit non-significant; Figure 4), while the addition of the micellar delivery technology pushed this effect into significance and resulted in a 2.6-fold increase in systemic exposure (further discussed below). The large interindividual variability observed, particularly in half-life and mean residence time, is consistent with known variability in polyphenol metabolism and highlights the need for larger studies incorporating metabolic phenotyping.

It should also be noted that in the LMC formulation, the excipients medium-chain triglycerides and lecithin may facilitate micelle formation and influence intestinal absorption [35,36]. Although this study was not designed to isolate the effects of individual excipients, their potential contribution to the observed improvements in systemic exposure cannot be excluded and warrants further mechanistic investigation.

Compared to previously published nanoformulations of chrysin, LMC demonstrated superior pharmacokinetic performance. For instance, Ting et al. reported a C_max_ of 21.5 ng/mL following administration of 500 mg of chrysin in an oil-in-water nanoemulsion, whereas our formulation achieved a mean C_max_ of 87.3 ng/mL with 469 mg of chrysin—a more than fourfold increase at a comparable dose [24]. Similarly, phytosomal and solid dispersion formulations have demonstrated modest improvements in permeability and dissolution but have not consistently yielded systemic exposure exceeding ~50 ng/mL [25,26]. These comparisons suggest that the synergistic combination of micellar delivery, quercetin-mediated metabolic inhibition, and optimized excipient architecture in LMC provides a significant bioavailability advantage over existing chrysin nanoformulations.

Chrysin is extensively metabolized to conjugated derivatives, with chrysin sulfate being reported as the predominant circulating metabolite, often exceeding parent chrysin concentrations by ~30-fold in vivo [22]. These conjugates are thought to have limited biological activity, suggesting that strategies which increase systemic exposure of intact chrysin may enhance its therapeutic potential. Although the present study did not directly measure chrysin metabolites, the observed increase in systemic exposure with LMC indicates that delivery strategies which improve intestinal permeability and absorption can substantially alter overall bioavailability. Micellar systems are known to enhance dispersion of lipophilic compounds, protect them from enzymatic conjugation, and potentially reduce efflux via transporters such as MRP2 and BCRP [31,32,50]. Furthermore, the non-significant trend toward a longer apparent half-life for LMC may indicate altered elimination kinetics or even partial enterohepatic recycling. Together, these findings provide mechanistic plausibility that micellar encapsulation contributed significantly to the improved pharmacokinetic profile observed in this study.

### 4.2. Caco-2 Permeability Results

The Caco-2 cell model is a well-established in vitro system for assessing the intestinal permeability of orally administered compounds [44,45]. In this study, UFC showed low permeability (P_app_ ≈ 2.22 × 10^−6^ cm/s), consistent with previous reports [51]. Co-formulation with quercetin and rutin enhanced permeability, with both the NMC and LMC formulations showing P_app_ values over an order of magnitude higher than UFC. LMC exhibited the highest permeability, exceeding unformulated chrysin by more than 10-fold.

These findings align with prior studies demonstrating that flavonoid co-formulation or encapsulation—via cyclodextrins, nanoemulsions, or phytosomes—can improve permeability and cellular uptake [24,51]. Quercetin and rutin likely contributed by modulating efflux transporter activity and enhancing paracellular absorption.

Interestingly, NMC showed improved in vitro permeability but the lowest systemic exposure in vivo. In contrast, LMC—formulated with the same flavonoid matrix—achieved the highest bioavailability. This highlights the critical role of micellar delivery vehicles. Beyond permeability enhancement, micelles likely improved solubility, protected against degradation, and facilitated transport across the gastrointestinal tract. These combined effects suggest a synergistic benefit of using both flavonoid co-formulation and micellar encapsulation.

Although in vitro models such as Caco-2 monolayers cannot fully replicate the complexity of human intestinal absorption—including metabolic conversion, gut microbiota interactions, and enterohepatic recycling—they provide valuable mechanistic insight into intestinal permeability and formulation performance. In this study, the micellar chrysin formulation (LMC) demonstrated enhanced permeability in Caco-2 assays, which is consistent with the significant increase in systemic exposure observed clinically (AUC_0-24_ and C_max_). These complementary findings support the predictive utility of in vitro permeability assays and underscore how micellar delivery systems can overcome the inherent absorption limitations of poorly soluble flavonoids such as chrysin [24,25,44,45,51,52,53,54]. Together, these mechanistic and clinical data provide convergent evidence that formulation strategies incorporating micellar technology can meaningfully enhance oral bioavailability.

### 4.3. Thirty-Day Safety Evaluation

Daily supplementation with LMC for 30 days was generally well tolerated, with no serious adverse events and no clinically meaningful changes in liver, kidney, or hematological parameters. Mild gastrointestinal discomfort occurred in one participant and resolved when taken with food, while another developed a mild rash that subsided after discontinuation, suggesting possible hypersensitivity, though the influence of other confounding factors cannot be ruled out. Blood chemistry values largely remained within clinical reference ranges, although small but statistically significant reductions in fasting glucose were observed in both males (5.9 ± 0.4 to 5.1 ± 0.7 mmol/L, *p* = 0.0369) and females (5.2 ± 0.6 to 4.2 ± 0.5 mmol/L, *p* = 0.0009), alongside minor changes in sodium and total CO_2_ in males. These changes remained within physiological limits and were not associated with changes in HbA1c or other metabolic markers. It is unclear whether the glucose reduction observed is attributable to chrysin itself, to quercetin—well documented for modest glucose-lowering effects in clinical trials—or to the combined action of all three flavonoids [55]. Chrysin has limited evidence for glycemic effects, while quercetin has shown improvements in glucose handling via AMPK activation and enhanced peripheral uptake, and rutin demonstrates similar effects in animal models [56,57,58,59]. Our study was not designed to isolate these contributions, and future trials incorporating single-compound comparators are warranted. Overall, these findings support the short-term safety and tolerability of the novel micellar chrysin–flavonoid combination while identifying glucose regulation as a physiological effect of interest.

### 4.4. Formulation Characterization and Stability

Appendix A provides physicochemical context to support the main pharmacokinetic and permeability findings in this work. It reports on the solubilized concentrations of the three chrysin formulations as well as the dispersion behavior as assessed by particle size analysis. The formulation-dependent differences in clinical exposure parallel the in vitro characterization trends observed in biorelevant media. The micellar formulation (LMC) generated higher apparent solubilized concentrations of chrysin (and co-flavonoids) in SGF and SIF under standardized conditions and produced distinct dispersion behavior in water relative to NMC and UFC. While laser diffraction reflects bulk dispersion or agglomeration and not nanoscale PDI, the combination of (i) greater assay-passable (dissolved + colloidally dispersed) material and (ii) favorable dispersion characteristics is consistent with the higher C_max_ and AUC observed clinically and with enhanced permeability in Caco-2 assays. These data do not constitute mechanistic proof; rather, they provide biorelevant context supporting the hypothesis that micellar encapsulation and excipient architecture increase the fraction available for absorption while maintaining stability across gastrointestinal environments. Future work will profile metabolite formation and directly quantify micelle integrity in bile-salt media to refine the mechanistic link.

### 4.5. Overall Implications and Future Directions

This study provides the first clinical evidence that a micellar chrysin–quercetin–rutin formulation (LMC) can overcome the poor bioavailability of native chrysin, achieving a ~2–3-fold increase in systemic exposure and enhanced intestinal permeability. These pharmacokinetic gains were achieved without compromising safety: 30 days of supplementation was well tolerated, with only mild, reversible adverse events and no clinically relevant changes in core biochemical markers. Small, but significant, reductions in fasting glucose were observed, consistent with the known glycemic effects of quercetin, but whether these changes reflect chrysin, quercetin, rutin, or their synergy remains unclear.

Experiments regarding formulation characterization such as physicochemical data on solubilization and dispersion behavior support the observed pharmacokinetic improvements. However, additional analytical work, including more detailed measurements of micelle size distribution, polydispersity, zeta potential, encapsulation efficiency, and in vitro release kinetics, may further refine mechanistic understanding.

Strengths of this work include its randomized, double-blind, crossover design, integration of mechanistic (e.g., Caco-2 permeability) and clinical pharmacokinetic data, and detailed safety assessment. However, the modest sample size, relatively short duration, and enrollment of only healthy, normal-weight adults, may limit generalizability to clinical populations. Also, the absence of metabolite profiling may restrict mechanistic interpretation, and while our in vitro permeability assays provided formulation-level comparisons, they did not isolate the individual contributions of quercetin, rutin, or the micellar vehicle. Future work should incorporate these reductionist approaches, explore dose–response relationships, and extend safety and efficacy assessments to target populations, such as individuals with metabolic or inflammatory disorders, to determine whether enhanced systemic delivery translates into meaningful clinical outcomes.

## 5. Conclusions

This study demonstrated the first clinical evidence that a micellar chrysin–quercetin–rutin formulation substantially achieved greater systemic exposure than both non-micellar and unformulated chrysin, with more than a twofold increase in AUC_0–24_ and C_max_ and evidence of improved intestinal permeability.

Daily use for thirty days was well tolerated, with only mild, reversible adverse events and no clinically meaningful safety concerns, even at higher systemic chrysin levels. Notably, modest reductions in fasting glucose were detected, aligning with the reported metabolic effects of quercetin, although the relative contribution of each flavonoid requires further clarification. Together, these findings establish proof of concept that micellar technology, when combined with flavonoid co-formulation, can overcome the intrinsic bioavailability limitations of chrysin and warrants future studies to define dose–response relationships, metabolite profiles, and long-term clinical efficacy in humans.

## Figures and Tables

**Figure 1 antioxidants-14-01313-f001:**
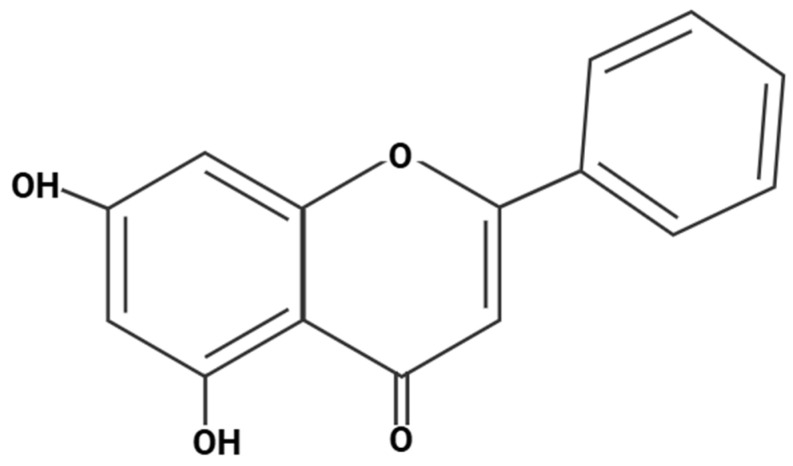
Chrysin chemical structure. Created in BioRender. Ibi, A. https://BioRender.com/3zkmz2i, accessed on 28 August 2025.

**Figure 2 antioxidants-14-01313-f002:**
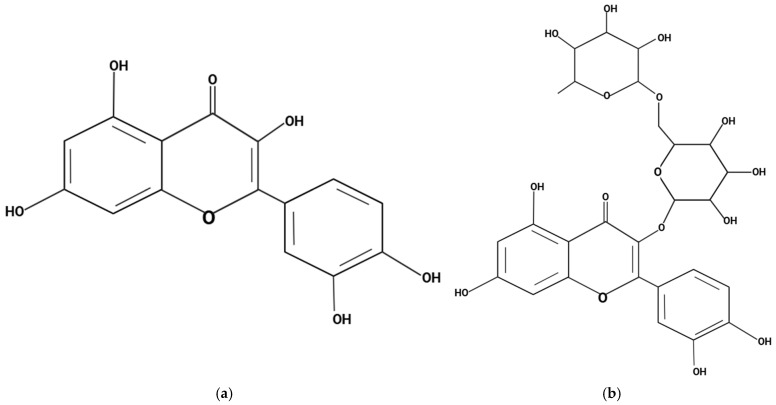
Chemical structures of (**a**) quercetin and (**b**) rutin. Created in BioRender. Ibi, A. https://BioRender.com/a75lxd7 and https://BioRender.com/nej4yyz accessed on 26 September 2025.

**Figure 3 antioxidants-14-01313-f003:**
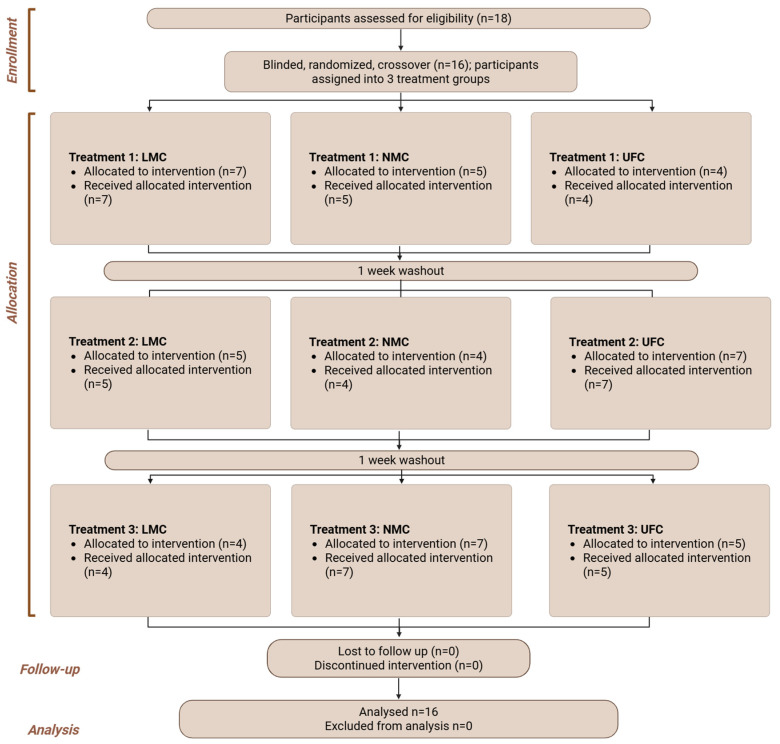
Flowchart of the study design showing the bioavailability phase (I) of the study. Created in BioRender. Ibi, A. https://BioRender.com/ohimwnn accessed on 28 August 2025.

**Figure 4 antioxidants-14-01313-f004:**
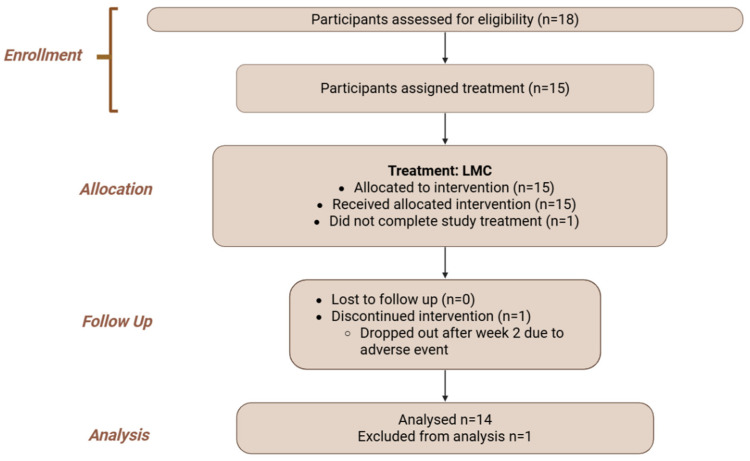
Flowchart of the study design showing the safety study phase (II) of the study. Created in BioRender. Ibi, A. https://BioRender.com/bsyxm6s accessed on 7 July 2025.

**Figure 5 antioxidants-14-01313-f005:**
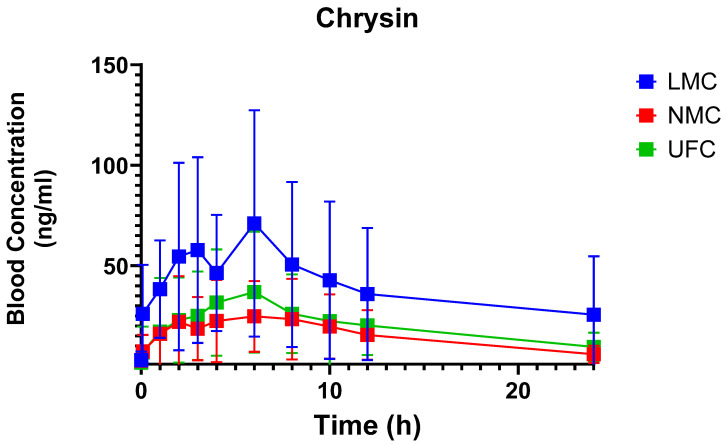
Mean plasma concentration–time profiles of chrysin following single oral administration of three formulations: micellar lipid matrix chrysin (LMC), non-micellar chrysin (NMC), and unformulated chrysin (UFC). Data are presented as mean ± SD (*n* = 13–16 per time point).

**Figure 6 antioxidants-14-01313-f006:**
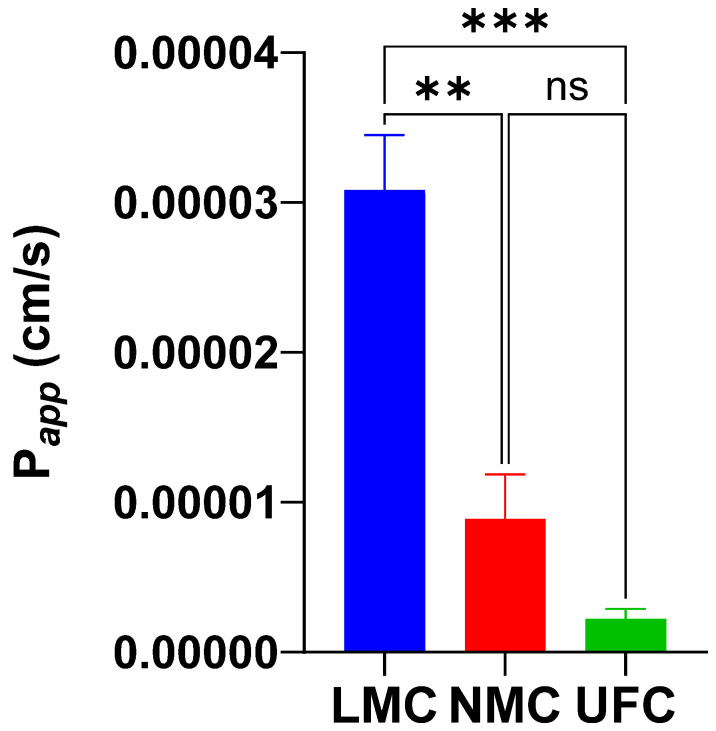
Bar graph showing the mean ± standard deviation of apparent permeability coefficients (P_app_) for three chrysin formulations—LMC, NMC, and UFC—across Caco-2 monolayers (*n* = 3 per group). The LMC formulation demonstrated significantly higher permeability than both NMC and UFC, suggesting enhanced potential for intestinal absorption. One-way ANOVA with Tukey post hoc correction. ns = not significant; ** *p* < 0.01; *** *p* < 0.001.

**Table 1 antioxidants-14-01313-t001:** Study interventions.

Intervention	LMC	NMC	UFC
Dosage form	Soft-gelatin capsules	Hard-gelatin capsules	Hard-gelatin capsules
Co-formulation	Quercetin, Rutin	Quercetin, Rutin	None
Micellar	Yes	No	No
Number of capsules per dose	1	1	1
Physical form of capsule content	Liquid	Powder	Powder
Chrysin (mg/dose)	469	412	558
Quercetin (mg/dose)	105	77.4	0
Rutin (mg/dose)	115	106.4	0
Lead (mg/kg)	<0.03	<0.03	<0.03
Mercury (mg/kg)	<0.02	0.029	<0.02
Cadmium (mg/kg)	<0.02	<0.02	<0.02
Arsenic (mg/kg)	<0.2	<0.2	<0.2
Excipients	Gelatin, glycerin, purified water, medium chain triglycerides, MSM, xylitol, lecithin, *Stevia rebaudiana* leaf extract, peppermint oil	Gelatin capsule	Cellulose capsule

**Table 2 antioxidants-14-01313-t002:** Demographic and baseline characteristics.

	Males	Females	Combined
N	7	11	18
Age (years)	39.4 ± 8.3	35.0 ± 12.1	37.2 ± 10.2
Weight (kg)	72.1 ± 7.4	58.8 ± 13.6	65.5 ± 10.5
Height (cm)	172.7 ± 5.1	162.8 ± 7.5	167.8 ± 6.3
BMI	24.2 ± 2.9	22.0 ± 3.5	23.1 ± 3.2

**Table 3 antioxidants-14-01313-t003:** Pharmacokinetic parameters of chrysin following single-dose administration of three formulations (mean ± SD; *n* = 16).

	LMC	NMC	UFC	*p* Value
AUC_0–24_ (ng/mL·h)	914.8 ± 697.5	345.7± 280.8	456.9 ± 356.4	0.0221
C_max_ (ng/mL)	87.3 ± 59.4	32.9 ± 23.5	42.1 ± 28.4	0.0128
T_1/2_ (h)	25.3 ± 59.1	9.00 ± 3.3	11.0 ± 3.5	0.6963
T_max_ (h)	4.8 ± 3.8	5.8 ± 3.9	4.2 ± 2.7	0.4903
MRTi (h)	38.4 ± 85.9	13.3 ± 4.2	17.2 ± 4.8	0.4207

Data are expressed as the mean ± standard deviation. One-way ANOVA with post hoc test (Dunnett Correction) were performed, and the *p*-values represent statistical significance where *p* < 0.05 indicates significant differences.

**Table 4 antioxidants-14-01313-t004:** Statistical analysis of safety markers in males following the chrysin treatment ^1,2,3^.

Male	Mean ± SD					*p*-Value	Normal Range
	Week 0	Week 1	Week 2	Week 3	30 Days		
Total Bilirubin (µmol/L)	14.1 ± 6.2	15.0 ± 8.8	16.1 ± 3.9	16.2 ± 5.5	19.7 ± 9.0	0.4701	3.4–21.0
AST (U/L)	24 ± 3.5	27.6 ± 10.5	23.2 ± 3.1	21.6 ± 2.6	24.6 ± 8.0	0.4799	15.0–40.0
ALT (U/L)	35.4 ± 17.2	36.6 ± 20.9	33.8 ± 15.7	30.6 ± 11.2	31.8 ± 7.8	0.5766	9.0–50.0
Creatinine (µmol/L)	73.4 ± 11.5	75.9 ± 14.5	79.4 ± 8.5	72.3 ± 4.8	80.9 ± 11.2	0.5761	44.0–97.0
eGFR (mL/min/1.73 m^2^)	110.5 ± 7.7	106.6 ± 11.8	105.4 ± 7.9	112.0 ± 5.1	103.6 ± 13.5	0.5746	>90
Glucose (mmol/L)	5.9 ± 0.4	5.9 ± 0.5	5.1 ± 0.3	5.3 ± 0.4	5.1 ± 0.7	0.0369 ^2^	3.89–6.11
HDL (mmol/L)	1.4 ± 0.2	1.3 ± 0.1	1.5 ± 0.2	1.3 ± 0.1	1.5 ± 0.2	0.0944	1.16–1.42
LDL (mmol/L)	4.3 ± 1.2	4.4 ± 1.4	4.2 ± 1.0	4.2 ± 0.8	4.4 ± 1.3	0.8209	0.50–3.14
TC (mmol/L)	6.5 ± 1.4	6.5 ± 1.6	6.4 ± 1.2	6.2 ± 0.8	6.7 ± 1.1	0.4223	0–5.17
TG (mmol/L)	1.6 ± 0.6	1.7 ± 0.9	1.4 ± 0.6	1.4 ± 0.4	1.8 ± 0.8	0.3436	0–1.70
HbA1c (mmol/L)	4.4 ± 0.4	4.4 ± 0.3	4.5 ± 0.2	6.4 ± 3.6	4.7 ± 0.4	0.3001	3.89–6.11
tCO2 (mmol/L)	22.4 ± 0.7	22.2 ± 1.0	22.9 ± 1.2	23.1 ± 1.5	25.0 ± 1.2	0.0118 ^3^	22.0–29.0
Ca (mmol/L)	2.6 ± 0.1	2.6 ± 0.06	2.6 ± 0.08	2.6 ± 0.06	2.5 ± 0.1	0.0865	2.00–2.58
P (mmol/L)	1.1 ± 0.1	1.0 ± 0.1	0.9 ± 0	1.1 ± 0.1	1.1 ± 0.1	0.0927	0.85–1.51
Mg (mmol/L)	0.9 ± 0.1	0.9 ± 0.1	0.9 ± 0.1	0.9 ± 0.05	0.9 ± 0.1	0.5788	0.65–1.25
K (mmol/L)	4.8 ± 0.3	4.8 ± 0.2	5.0 ± 0.2	4.6 ± 0.4	4.6 ± 0.3	0.3732	3.40–5.30
Na (mmol/L)	140.2 ± 1.9	140.1 ± 1.9	139.6 ± 3.1	138.5 ± 3.4	133.8 ± 0.7	0.0061 ^3^	135.0–147.0
Cl (mmol/L)	105.0 ± 1.5	106.0 ± 2.3	104.5 ± 5.6	105.0 ± 1.0	103.3 ± 3.1	0.5246	99.0–122.0

^1^ Data are expressed as the mean ± standard deviation. One-way ANOVA with post hoc test (Dunnett Correction) were performed, and the *p*-values represent statistical significance across the 5 measurement time points, where *p* < 0.05 indicates significant differences. ^2^ A statistically significant effect of time over the 30-day period was observed, but post hoc comparisons using Dunnett’s test showed no significant differences between baseline (Day 0) and any subsequent time point (all *p* > 0.05). ^3^ A statistically significant effect of time over the 30-day period was observed. Post hoc comparisons using Dunnett’s test also showed a significant change between baseline (Day 0) and Day 30 (*p* < 0.05). No other time points differed significantly from baseline.

**Table 5 antioxidants-14-01313-t005:** Statistical analysis of safety markers in females following the chrysin treatment ^1,2^.

Female	Mean ± SD					*p*-Value	Normal Range
	Week 0	Week 1	Week 2	Week 3	30 Days		
Total Bilirubin (µmol/L)	12.4 ± 8.9	10.7 ± 7.9	17.1 ± 7.2	11.0 ± 4.1	10.3 ± 5.7	0.2388	3.4–21.0
AST (U/L)	19.6 ± 5.7	20.1 ± 4.5	20.8 ± 5.2	19.3 ± 3.7	19.8 ± 3.6	0.8371	13.0–35.0
ALT (U/L)	18.8 ± 4.7	19.9 ± 6.1	20.6 ± 10.6	18.7 ± 4.4	19 ± 1.5	0.6655	7.0–40.0
Creatinine (µmol/L)	57.4 ± 8.5	55.2 ± 10.9	63.2 ± 9.6	64.8 ± 10.6	58.8 ± 8.9	0.116	35.0–80.0
eGFR (mL/min/1.73 m^2^)	115.6 ± 12.6	114.8 ± 12.4	109.1 ± 15.5	106.0 ± 15.7	114.1 ± 13.6	0.16	>90
Glucose (mmol/L)	5.2 ± 0.6	5.3 ± 0.5	5.1 ± 0.8	4.9 ± 0.7	4.2 ± 0.5	0.0009 ^2^	3.89–6.11
HDL (mmol/L)	1.6 ± 0.3	1.7 ± 0.3	1.6 ± 0.3	1.6 ± 0.3	1.6 ± 0.3	0.3011	1.29–1.55
LDL (mmol/L)	3.3 ± 1.0	3.5 ± 1.2	3.4 ± 1.1	3.4 ± 1.2	3.3 ± 1.0	0.5575	0.50–3.14
TC (mmol/L)	5.4 ± 1.1	5.6 ± 1.3	5.5 ± 1.2	5.4 ± 1.4	5.3 ± 1.2	0.3943	0–5.17
TG (mmol/L)	1.3 ± 0.5	1.0 ± 0.3	1.2 ± 0.7	1.1 ± 0.3	1.0 ± 0.3	0.3549	0–1.70
HbA1c (mmol/L)	4.4 ± 0.4	4.4 ± 0.5	4.4 ± 0.6	4.6 ± 0.5	4.3 ± 0.4	0.4551	3.89–6.11
tCO2 (mmol/L)	22.2 ± 0.6	23.2 ± 3.1	22.4 ± 1.1	23.1 ± 2.1	24.5 ± 2.0	0.1702	22.0–29.0
Ca (mmol/L)	2.4 ± 0.52	2.2 ± 0.59	2.5 ± 0.08	2.53 ± 0.08	2.4 ± 0.23	0.3504	2.00–2.58
P (mmol/L)	1.2 ± 0.1	1.3 ± 0.5	1.1 ± 0.09	1.2 ± 0.07	1.2 ± 0.2	0.4523	0.85–1.51
Mg (mmol/L)	0.9 ± 0.07	1.0 ± 0.4	0.9 ± 0.1	0.9 ± 0.03	0.9 ± 0.1	0.3011	0.65–1.25
K (mmol/L)	4.8 ± 0.2	4.6 ± 0.2	4.9 ± 0.2	4.5 ± 0.3	4.5 ± 0.6	0.115	3.40–5.30
Na (mmol/L)	140.6 ± 2.9	134.5 ± 15.4	139.7 ± 3.1	138.3 ± 2.8	133.4 ± 8.0	0.2726	135.0–147.0
Cl (mmol/L)	105.8 ± 3.9	103.8 ± 4.5	106.2 ± 3.2	105.8 ± 1.6	102.1 ± 7.0	0.2791	99.0–122.0

^1^ Data are expressed as the mean ± standard deviation. One-way ANOVA with post hoc test (Dunnett Correction) were performed, and the *p*-values represent statistical significance across the 5 measurement time points, where *p* < 0.05 indicates significant differences. ^2^ A statistically significant effect of time over the 30-day period was observed. Post hoc comparisons using Dunnett’s test also showed a significant change between baseline (Day 0) and Day 30 (*p* < 0.05). No other time points differed significantly from baseline.

## Data Availability

The original contributions presented in this study are included in the article. Further inquiries can be directed to the corresponding authors.

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
