# Peer review of "Comparative Pharmacokinetics and Safety of a Micellar Chrysin–Quercetin–Rutin Formulation: A Randomized Crossover Trial"

_antioxidants, 2025, doi:10.3390/antiox14111313_

Round 1
Reviewer 1 Report
The authors investigated the effect on the pharmacokinetics and safety of the method of administration of the flavonoid chrysin to healthy volunteers. The flavonoid was administered in the form of a novel formulation co-encapsulated with quercetin and rutin, as a non-micellar formulation, and as an unformulated form. The study (a double-blind, randomized, three-period crossover trial) showed that the new formulation significantly improves the bioavailability of chrysin, which is very poorly soluble in water, and is well tolerated by the human body.
The work is interesting, and the design of the study, the way it was performed, and the discussion of the results obtained do not raise any objections. The cited literature is up-to-date and closely related to the problem under study. A large part of the cited articles have been published in the last ten years. Before publishing the work, the authors should clarify a few issues.
The Introduction should include a figure showing the structural formula of the studied flavonoid chrysin and provide a little more information about its physicochemical properties. Chrysin is poorly soluble in water, but are there any known factors that can increase its solubility in water, apart from lipid-based delivery systems, such as emulsions, phytosomes, and solid dispersions?
Figure 1 can be simplified. If Allocated to interventions (n=7) and Received allocated intervention (n=7), is it necessary to write Did not receive allocated intervention/absent on treatment day (n=0)?
Table 1, Table 4, and Table 5 – the concentration units should be standardized. In Table 1, the concentrations of Mercury, Cadmium, and Arsenic are given in ppm, which are not SI units, while in Tables 4 and 5, the concentrations of Ca, P, Mg, K, Na, and Cl are given in mmol/L.
Figure 3, Table 3, and text should also use a consistent notation for units: ng/mL or ng mL–1.
References need to be adapted to the requirements of the journal Antioxidants, including the use of appropriate journal name abbreviations. In addition, refs. 29 and 32 lack complete bibliographic data.
Reviewer 2 Report
Dear author, the thematic reported in this study is quite interesting. However the paper suffers of a major limit: the lack of data concerning the development and optimisation of the formulation and its characterisationto obtain reproducible results. No data of selection of the exicipients, flavonoids' recovery and loading percentage, micelle sizes, PDI and zeta potential, chemical and physical stability in gastric and enteric media, release properties etc..
Dear authors,
some corrections are needed to improve the present study in addition to the part dedicated to the formulation.
Title: the entire coformulation should be reported.
Introduction: literatures concerning all other formulations LipoMicel should be added and discussed. I reported below some of these papers
Experimental: to evaluate the effects of rutin and curcumin and in addition of the unloaded formulation should be tested at least in vitro with caco2 cells to evaluate the possible contribution of each of them.
The superiority of the present formulation by comparison with other published nanoformulations should be reported
Reviewer 3 Report
The manuscript of the article prepared by Afoke Ibi, Chuck Chang, Yun Chai Kuo, Yiming Zhang, Peony Do, Min Du, Yoon Seok Roh, Roland Gahler, Mary Hardy and Julia Solnier “Comparative Pharmacokinetics, Bioavailability and Safety Evaluation of a Novel Micellar Chrysin-Flavonoid Formulation: Results from a Double-Blind, Randomized Crossover Clinical Trial” described studies of new micellar formulation of chrysin combined with quercetin and rutin (LMC) to improve chrysin’s poor oral bioavailability.
The manuscript is well-written and would be interesting for the experts of related fields, however, some issues needed to be added and addressed before publication.
- Please specify the exact number of people who participated in the study, the introduction says 16, the results section already mentions 18.
- In experimental part please give brief description of procedures for micelle formulation preparation and characterisation.
- Why amount of active ingredients in the all 3 different formulations are not the same (data from Table 1)?
- Shouldn't a study also be conducted on a sample containing only quercetin and rutin without any chrysin?
- Why were such a wide age range chosen for both genders? Is it correct to calculate the average age? Maybe the results can be divided into age groups somehow? However, there is probably not enough participants for that.
- What is meant by the term "Normal Range" in tables 4 and 5? Why are these data identical in both tables, does this mean that there is no data for each gender separately?
- Please rewrite conclusions not so general way.
Consequently, I do recommend accepting this manuscript for publication with major revision.
See paragraph - major comments
Round 2
Reviewer 2 Report
Dear Authors the manuscript has the main concerns reported in the first revision.
All the critical issues indentified in the first revision have not been effectively addressed
Reviewer 3 Report
The main recommendations have been addressed and the necessary corrections implemented. It may be worth considering whether the introduction could be shortened. Additionally, presenting the compound structures in the same format would be preferable.
Please, see paragraph Major comments
